# Modified Curcuminoid-Rich Extract Liposomal CRE-SDInhibits Osteoclastogenesis via the Canonical NF-κB Signaling Pathway

**DOI:** 10.3390/pharmaceutics15092248

**Published:** 2023-08-30

**Authors:** Sompot Jantarawong, Piyawut Swangphon, Natda Lauterbach, Pharkphoom Panichayupakaranant, Yutthana Pengjam

**Affiliations:** 1Division of Biological Science, Faculty of Science, Prince of Songkla University, Songkhla 90110, Thailand; sompot.jant@gmail.com; 2Faculty of Medical Technology, Prince of Songkla University, Songkhla 90110, Thailand; piyawut.s@psu.ac.th (P.S.); natda.t@psu.ac.th (N.L.); 3Department of Pharmacognosy and Pharmaceutical Botany, Faculty of Pharmaceutical Sciences, Prince of Songkla University, Songkhla 90110, Thailand; pharkphoom.p@psu.ac.th; 4Phytomedicine and Pharmaceutical Biotechnology Excellence Center, Faculty of Pharmaceutical Sciences, Prince of Songkla University, Songkhla 90110, Thailand

**Keywords:** curcuminoid, liposomal CRE-SD, osteoclastogenesis, RANKL-stimulated RAW 264.7 macrophage, canonical NF-κB signaling pathway

## Abstract

Curcuminoids, namely curcumin, demethoxycurcumin, and bisdemethoxycurcumin, are the major active compounds found in *Curcuma longa* L. (turmeric). Although their suppressive effects on bone resorption have been demonstrated, their pharmacokinetic disadvantages remain a concern. Herein, we utilized solid dispersion of a curcuminoid-rich extract (CRE), comprising such curcuminoids, to prepare CRE-SD; subsequently, we performed liposome encapsulation of the CRE-SD to yield liposomal CRE-SD. In vitro release assessment revealed that a lower cumulative mass percentage of CRE-SD was released from liposomal CRE-SD than from CRE-SD samples. After culture of murine RANKL-stimulated RAW 264.7 macrophages, our in vitro examinations confirmed that liposomal CRE-SD may impede osteoclastogenesis by suppressing p65 and IκBα phosphorylation, together with nuclear translocation and transcriptional activity of phosphorylated p65. Blind docking simulations showed the high binding affinity between curcuminoids and the IκBα/p50/p65 protein complex, along with many intermolecular interactions, which corroborated our in vitro findings. Therefore, liposomal CRE-SD can inhibit osteoclastogenesis via the canonical NF-κB signaling pathway, suggesting its pharmacological potential for treating bone diseases with excessive osteoclastogenesis.

## 1. Introduction

Bone homeostasis dynamically regulates the coupling of bone formation and bone resorption. Osteoclasts are large bone breakdown cells that function in collaboration with osteoblasts (bone-forming cells), osteocytes, and other cells in the bone homeostasis mechanism. Osteoclastogenesis (osteoclast differentiation) can be stimulated by the specific binding between the receptor activator of nuclear factor-κB ligand (RANKL) and the receptor activator of nuclear factor-κB (RANK) of the RAW 264.7 macrophages—the osteoclast precursors [1,2,3]. After osteoclastogenesis is complete, multinucleated microscopic appearance can be detected, as well as an increased expression of osteoclast markers (e.g., cathepsin K (CTSK), tartrate-resistant acid phosphatase (TRAP), c-Fos, and the nuclear factors of activated T cells 1 (NFATc1) [4,5]. Many previous reports have revealed that the immoderate osteoclast activity, surpassing bone resorption, leads to an imbalance in bone homeostasis and, subsequently, results in some types of bone diseases, such as osteoporosis and rheumatoid arthritis (RA) [6,7]. This has contributed to the design of drugs impeding the excessive osteoclast action that occurs during the treatment of bone diseases. To achieve this, experimental (in vitro and in vivo) and computational or bioinformatics (in silico) analyses that elucidate the biochemical effects of drugs on osteoclasts are required. The inhibitory effects of various anti-osteoporotic and antirheumatic synthetic drugs on osteoclast activities have been explained [8,9]. Furthermore, an in silico experimental method known as molecular docking was used to demonstrate that some established drugs, such as CTSK inhibitors and interleukin 6 inhibitors, can interact with protein targets to hamper osteoclast differentiation and alleviate bone diseases [10,11]. Despite the benefits in pharmaceutical practice, the serious side effects of synthetic drugs for the treatment of osteoporosis and RA are of concern [10,12]. Hence, advances in herbal extracts and an examination of the molecular roles of herbal extracts are alternative approaches that may lead to effective treatments for bone diseases.

*Curcuma longa* L. (turmeric) is a rhizomatous perennial in the Zingiberaceae (ginger) family planted in tropical and subtropical regions worldwide. It is widely utilized as an herbal medicine, owing to its manifold pharmacological properties, including antibacterial, anticancer, antioxidant, anti-inflammatory, and antiproliferative effects [13,14,15]. Curcuminoids—the main active ingredients in turmeric—are phenolic compounds commonly extracted for the aforementioned medicinal uses. The US Food and Drug Administration has classified these compounds as “Generally Recognized as Safe” (GRAS) [13,14]. The curcuminoid termed curcumin (Cu) is present in the highest proportion (approximately 80%), followed by demethoxycurcumin (De), and then by bisdemethoxycurcumin (Bis) [15,16]. The three-dimensional (3D) chemical structures of each curcuminoid are illustrated in Figure 1A–C. Numerous in vitro and in vivo studies, together with clinical trials, have shown that each of these curcuminoids can inhibit bone resorption and may be promising remedies for bone diseases [17,18,19]. Huang et al. [20] demonstrated the synergistic effects of typical treatment for HOS cancer cells in vitro with a combination of Cu, De, and Bis, compared with a combination containing only one or two of the curcuminoids. This finding suggests the prospect of employing such an herbal combination for osteosarcoma therapy. Although these biomedical examinations have corroborated the potential of curcuminoids in the management of bone diseases, curcumin pharmacokinetics are not ideal, as there are still many issues to consider, namely: restricted bioavailability, limited absorption in the small intestine, fast metabolism, and fast systemic elimination. Additionally, curcumin is hydrophobic and has low solubility in water, as it contains *o*-methoxy and phenolic groups [14,21,22]. In an effort to address these issues, various techniques through which to improve the pharmacokinetic attributes of curcumin have been applied; examples include preparing curcumin as a solid dispersion (dispersing the hydrophobic molecule in an inert carrier such as a synthetic polymer) and using liposomal encapsulation [22]. The superior water solubility and absorption properties of solid dispersion-based curcumin and liposomal curcumin compared with pure curcumin have been revealed in some recent studies [23,24]. However, these pharmacological attributes have not yet been proven for the combination of Cu, De, and Bis modified by integrating bothabovementioned formulation techniques. Moreover, the biomolecular regulatory effects of these newly developed forms of curcuminoids on osteoclast differentiation and bone resorption are yet to be elucidated.

The downstream signaling pathways of the nuclear factor kappa-light-chain-enhancer of activated B cells (NF-κB) are crucial for osteoclastogenic stimulation. In the canonical (classical) NF-κB signaling pathway, p50 (originating from the proteolysis of the p105 precursor protein) and p65 (Rel A) serve as transcription factors. When RANKL interacts with RANK on the surface of osteoclast precursors (in a mechanism called the RANKL–RANK upstream signaling pathway), the cytoplasmic adaptor molecules in the canonical NF-κB signaling pathway are subsequently activated. Then, the cytoplasmic inhibitor of the IκB kinase (IKK) complex, comprising two catalytic kinase subunits (IKKα and IKKβ) and at least one noncatalytic regulatory subunit known as the NF-κB essential modifier (NEMO), or IKKγ, regulates the phosphorylation of a specific subunit in the IκB inhibitor α (IκBα)/p50/p65 protein complex. The 3D structure of the IκBα/p50/p65 complex is shown in Figure 1D. Activated IKKβ phosphorylates IκBα, whereas IKKα phosphorylates p65. Afterward, the phosphorylated IκBα (p-IκBα) is polyubiquitinated and degraded by the 26S proteasome, whereas the p50/p65 heterodimer—the most structurally stable form of the NF-κB dimer—freely translocates to the nucleus to transcribe its target genes [25,26,27,28,29,30]. Previous in vitro experiments revealed that curcumin can hinder the phosphorylation and degradation of IκBα, as well as the nuclear translocation of p65 [31]. By applying curcumin as a cell treatment, the phosphorylation of IκBα and p65 in osteoclasts, namely RANKL-stimulated RAW 264.7 macrophages, was suppressed, which eventually suppressed osteoclastogenesis [32,33]. In molecular docking simulations, Cheemanapalli et al. used AutoDock version 4.0 to show that there were binding interactions between curcumin and the specific active site residues of each major protein in the canonical NF-κB pathway, especially the IKK complex, proteasome, and p65 [34]. In addition, Saeed et al. demonstrated strong binding affinity between the IκBα/p50/p65 complex and diverse curcumin compounds (including Cu, De, Bis, etc.) using AutoDock version 4.2.6 [35].

Although several in vitro and bioinformatics investigations have evidenced the protective effect of curcumin on the canonical NF-κB pathway in osteoclast differentiation, the biomolecular mechanism has not been fully explained for the combination of Cu, De, and Bis modified by utilizing both solid dispersion and liposomal encapsulation techniques. To fill these knowledge gaps, we prepared a curcuminoid-rich extract (CRE), containing Cu, De, and Bis from the dried powders of *C. longa* rhizomes, and employed the synthetic polymer polyvinylpyrrolidone K30 (PVP K30) to produce a solid dispersion, as illustrated in Figure 2. After obtaining the CRE with PVP K30 solid dispersion (CRE-SD) from this process, it was encapsulated in liposomes to derive liposomal CRE-SD. Subsequently, we evaluated the characteristics of the liposomal CRE-SD and performed an in vitro analysis of its biomolecular involvement in the canonical NF-κB pathway of murine RANKL-stimulated RAW 264.7 macrophages. Moreover, to illustrate the molecular interactions, we implemented a blind docking simulation between the IκBα/p50/p65 complex and each of the curcuminoids, using the web-based molecular docking tool CB-Dock2.

## 2. Materials and Methods

### 2.1. In Vitro Experiments

#### 2.1.1. Cells and Reagents

Murine RAW 264.7 macrophages were purchased from the ATCC Cell Bank (Biomedia, Bangkok, Thailand). RANKL was purchased from R&D Systems (Minneapolis, MN, USA). RPMI 1640 cell culture medium was purchased from Gibco Co. (Bangkok, Thailand). The RNeasy kit for RNA extraction was purchased from Qiagen (Valencia, CA, USA). The other chemicals mentioned in the study were purchased from Sigma Chemical Co. (Bangkok, Thailand).

#### 2.1.2. Preparation, Characterization, and In Vitro Release of Liposomal CRE-SD

CRE containing 88% *w*/*w* total curcuminoids (72.81 ± 0.83% *w*/*w* Cu, 12.49 ± 0.57% *w*/*w* De, and 4.24 ± 0.16% *w*/*w* Bis; the curcuminoid content was measured by Lateh et al. using HPLC analysis [36]) and CRE-SD containing 7% *w*/*w* curcuminoids were prepared using the method described by Lateh et al. [36,37]. The dried powder of *C. longa* rhizomes was dissolved in ethanol, and microwave-assisted extraction was performed (900 W and three irradiation cycles; each cycle comprised 3 min on followed by 30 s off). Then, the filtration and elution of the resulting *C. longa* extracts were conducted to derive CRE. For the latter process, a Diaion^®^ HP-20 column (Mitsubishi Chemical Co., Tokyo, Japan) with the incorporation of 55% and 60% *v*/*v* ethanol was utilized. To obtain CRE-SD, PVP K30 (10% mass ratio) was mixed with CRE, and solvent evaporation of the CRE mixture was performed under reduced pressure.

The liposomal suspension (25 mol 1-palmitoyl-2-oleoyl-phosphatidylcholine (POPC):1 mol CRE-SD in phosphate-buffered saline (PBS); pH 7.4) was prepared in accordance with the procedure reported by Sinjari et al. [38]. A POPC aliquot (Avanti Polar Lipids, Alabama, USA) was first dissolved in chloroform. Then, the aliquot was transferred to a round-bottomed flask and dried in a rotary evaporator (low pressure, 40 °C). A phospholipid film formed on the inner wall of the flask. This film was stored overnight at 4 °C, rehydrated with PBS buffer (pH 7.4), and sonicated for 30 min. The resulting liposome suspensions were sterilized for 2 h under a UV lamp. Finally, an appropriate quantity of CRE-SD, dissolved in dimethylsulfoxide, was added to the suspensions.

To evaluate the characteristics of the liposomes, i.e., the liposome hydrodynamic diameter, polydispersity index (PDI), and zeta potential at pH 7.4, we applied the procedure described by Pengjam et al. [39]. We measured the liposomal diameter and PDI using dynamic light scattering (DLS) via a Zetasizer Nano-S instrument (Malvern Instruments, Malvern, UK). To derive the surface charge of the liposomes, their electrophoretic mobility (*μ*) was measured using correlation spectroscopy via a Zetasizer Nano-S instrument. We used the Smoluchowski equation, *Z* = *μη*/*ε*, to calculate the zeta potential (*Z*), where *η* and *ε* indicate viscosity and solution permittivity, respectively. All measurements were performed at room temperature, and 200 μL of each sample was diluted 20-fold in a citric–phosphate buffer solution.

We assessed the in vitro release of CRE-SD using the method stated by Pengjam et al. [39]. The overnight immersion of dialysis tubing (molecular weight cutoff: 12,000–14,000 Da; Carolina Biological Supply Company (Burlington, NC, USA)) was performed in deionized water to remove preservatives. We employed a dialysis sac with CRE-SD and another dialysis sac with liposomal CRE-SD. After the dialysis sacs were sealed, PBS (pH 7.4) was dissolved in 25% *v*/*v* methanol in a flask using a glass stirring rod. Then, the flask, maintained at 37.0 ± 0.1 °C, was stirred constantly with a magnetic stirrer at 150 rpm. The mass of CRE-SD leaked from each sac was measured every 10 days, from day 0 to day 70. Then, the cumulative mass percentage of CRE-SD released at each timepoint was derived using the following formula:(1)$$\mathrm{Percentage}\;\mathrm{of}\;\mathrm{cumulative}\;\mathrm{mass}\;\mathrm{of}\;\mathrm{released}\;\mathrm{CRE}-\mathrm{SD}\;\mathrm{on}\;\mathrm{day}\;(10\times (n-\;1))=\frac{\sum_{i=1}^{n}m_{i}}{M_{CRE-SD}}\;\times \;100$$
where *n* represents the timepoint of the measurement, ranging from 1 to 8, i.e., days 0, 10, …, 70, respectively; $\sum_{i=1}^{n}m_{i}$ denotes the summation of the masses of the leaked CRE-SD from timepoint 1 to timepoint *n*; and *M_CRE-SD_* indicates the mass of the CRE-SD incorporated to the dialysis sac on day 0.

#### 2.1.3. Culture of RAW 264.7 Cells

RAW 264.7 cells were grown in RPMI 1640 medium, supplemented with 10% heat-immobilized fetal bovine serum (FBS), 100 U/mL penicillin G, 2 mM glutamine, and 100 µg/mL streptomycin sulfate. The cells were incubated in a cell culture vessel at 37 °C under humid conditions and 5% carbon dioxide. At the subconfluent stage, when approximately 70–80% of the surface of the cell culture vessel was occupied, the cells were harvested, following the application of 0.25% trypsin and 0.02% ethylenediaminetetraacetic acid. Then, 5 × 10^3^ RAW 264.7 cells were seeded into each well of a culture plate for the subsequent in vitro experiments. The culture medium (RPMI 1640 containing supplements) was replenished every 3 days.

#### 2.1.4. Evaluation of Optimal RANKL Incubation Time in RAW 264.7 Cells Treated with CRE-SD-Free POPC Liposomes (CRE-SD-FREE-LIP)

To investigate the optimal RANKL incubation time, RAW 264.7 cells with and without 20 ng/mL RANKL treatments were incubated with CRE-SD-FREE-LIP for 1, 2, 3, 4, and 5 days. A TRAP assay was performed to detect TRAP activity. The TRAP assay kit was purchased from TaKaRa Bio Inc., Tokyo, Japan. The relative expression of CTSK was measured using the qRT–PCR procedure described by Pengjam et al. [39].

#### 2.1.5. Viability Examination of Liposomal CRE-SD-Treated RAW 264.7 Cells

RAW 264.7 cells were classified into two groups: RANKL-stimulated RAW 264.7 cells, which were subjected to 20 ng/mL RANKL for 5 days; and RAW 264.7 cells without RANKL stimulation. CRE-SD-FREE-LIP and liposomal CRE-SD (1, 2.5, 5, 10, 20, and 30 µg/mL) were added to the cultured cells. Thereafter, an MTT assay was conducted to examine viable cells. The absorbance of each cell sample at 570 nm was measured using a Thermo Scientific spectrophotometer (Multiskan FC, Pittsburgh, PA, USA). Thereafter, the resulting absorbance was converted into the number of viable cells.

#### 2.1.6. Confirmation of Osteoclastogenic Inhibition in Liposomal CRE-SD-Treated RANKL-Stimulated RAW 264.7 Cells

For the TRAP staining assay, RANKL-unstimulated RAW 264.7 cells, along with 20 ng/mL RANKL-stimulated RAW 264.7 cells with CRE-SD-FREE-LIP and 20 µg/mL liposomal CRE-SD, were incubated for 5 days in a cell culture vessel under the previously described conditions. On day 5, TRAP staining was performed, and the morphology of the cells was subsequently observed under a brightfield light microscope at 1000× magnification. Purplish to dark red granules in the cytoplasm denoted the presence of intracellular TRAP. The percentage of multinucleated TRAP-positive cells, i.e., the cells containing stained granules, was calculated.

To measure TRAP activity, 20 ng/mL RANKL-stimulated RAW 264.7 cells were incubated with CRE-SD-FREE-LIP and liposomal CRE-SD (5, 10, and 20 µg/mL) for 5 days. After cell culture, the TRAP activity was measured using the TRAP assay kit, as previously described.

To examine F4/80 expression, RANKL-unstimulated RAW 264.7 cells, along with 20 ng/mL RANKL-stimulated RAW 264.7 cells with CRE-SD-FREE-LIP and 20 µg/mL liposomal CRE-SD, were incubated for 5 days under the previously described cell culture conditions. As the anti-F4/80 antibody can interact with the F4/80 antigen, a murine macrophage marker [40], immunocytochemical staining for anti-mouse F4/80 antibody, using method reported by Pengjam et al. [39], was performed to analyze the expression of F4/80, which was representative of the abundance of murine macrophages in each sample. The following steps were conducted at room temperature. Each cell sample was rinsed in PBS and subsequently fixed for 10 min in 4% paraformaldehyde dissolved in PBS. Subsequently, 0.3% hydrogen peroxide in absolute methanol was added to the samples for 15 min to inhibit the cellular endogenous peroxidase activity. Then, the samples were preincubated for 1 h with 500 μg/mL normal goat IgG antibody dissolved in 1% bovine serum albumin (BSA) in PBS (pH 7.4). The samples were incubated for 2 h with anti-mouse F4/80 antibody, washed with 0.075% Brij 35 in PBS, and then incubated for 1 h with horseradish peroxidase (HRP)-conjugated goat anti-mouse IgG antibody in 1% BSA in PBS. They were further washed with 0.075% Brij 35 in PBS. Then, under dark conditions, the cells were exposed to the substrate 3,3′-diaminobenzidine (DAB), along with a nickel and cobalt solution (DAB enhancer). Regarding the reaction catalyzed by the HRP, the DAB oxidized by hydrogen peroxide resulted in the formation of brown patches. Finally, the cell samples were mounted onto glass slides, and microscopic images were captured using an Olympus brightfield light microscope and a digital camera. At 1000× magnification, brown patches in the microscopic images indicate stained murine macrophages, hereafter referred to as F4/80-positively stained cells. Additionally, the percentage of F4/80-positively stained cells was computed. RAW 264.7 cells treated with CRE-SD-FREE-LIP without RANKL stimulation were used as the control.

To detect the expression of osteoclast marker genes, 20 ng/mL RANKL-treated RAW 264.7 cells exposed to CRE-SD-FREE-LIP and liposomal CRE-SD (10 and 20 µg/mL) were incubated for 5 days under the previously described cell culture conditions. Then, the relative expressions of the CTSK, c-Fos, and NFATc1 genes were measured using RT–PCR. Total RNA was extracted using an RNeasy Mini Kit (Qiagen, USA), and complementary DNA was synthesized using a ReverTra Ace qPCR kit (Toyobo, Osaka, Japan). A Fast-Start SYBR Green Master Mix and a BIOER LifeECO™ PCR Thermal Cycler (Bangkok, Thailand) were utilized for RT-PCR amplification. The amplicons were then transferred to a Mupid-EXU Gel Electrophoresis System (Bangkok, Thailand) for agarose gel electrophoresis. Images of the gels were captured using a UVITEC Gel Documentation System (Bangkok, Thailand) and analyzed using a Bio-Rad ChemiDoc MP Imaging System (Bangkok, Thailand).

#### 2.1.7. Primer Sequences for RT–PCR and qRT–PCR Amplification

The primer sequences used for RT-PCR and qRT-PCR are shown below:
CTSK Forward: ATGTGGGGGCTCAAGGTTCTG Reverse: CATATGGGAAAGCATCTTCAGAGTCc-Fos Forward: CCAGTCAAGAGCATCAGCAA Reverse: AAGTAGTGCAGCCCGGAGTANFATc1 Forward: CCGTTGCTTCCAGAAAATAACA
 Reverse: TGTGGGATGTGAACTCGGAAGAPDH (internal control)
 Forward: AAATGGTGAAGGTCGGTGTG
 Reverse: GAATTTGCCGTGAGTGGAGT

#### 2.1.8. Detection of Phosphorylation of p65 and Iκbα Proteins in Liposomal CRE-SD-Treated RANKL-Stimulated RAW 264.7 Cells

Western blotting was used to detect the phosphorylation of p65 and IκBα. First, RAW 264.7 cells stimulated with 20 ng/mL RANKL were exposed to CRE-SD-FREE-LIP and liposomal CRE–SD (10 and 20 µg/mL) for 5 days. Cytoplasmic and nuclear protein extraction were performed using the NE–PER reagent (Thermo Scientific Inc., Washington, DC, USA) in accordance with the manufacturer’s protocol. Subsequently, SDS–PAGE was used to separate proteins. Then, the separated proteins were blotted onto polyvinyl difluoride membranes. Nonspecific binding to the membrane was blocked prior to incubation with the following primary antibodies, which were purchased from Cell Signaling Technology (Beverly, MA, USA): anti-p65, anti-phosphorylated p65, anti-IκBα, anti-phosphorylated IκBα, and anti-β-actin antibodies. The membrane was washed to eliminate unbound antibodies, and then incubated for 1 h with HRP-conjugated secondary antibodies. The membrane was washed again. Finally, a Bio-Rad ChemiDoc MP Imaging System (Bangkok, Thailand) was employed to detect the protein bands on the membrane.

#### 2.1.9. Confirmation of the Inhibitory Effect of Liposomal CRE-SD on Nuclear Translocation and Transcriptional Activity of Phosphorylated p65 (p-p65)

To confirm whether liposomal CRE-SD impedes nuclear translocation and the transcriptional activity of p-p65, subconfluent RAW 264.7 cells were treated for 4 h with 20 ng/mL RANKL and JSH23 (4-methyl-*N*1-(3-phenylpropyl)-1,2-benzenediamine; IC_50_ = 7.1 μM), an inhibitor of NF-κB nuclear translocation and transcriptional activity. The cells were then incubated for 16 h with 20 μg/mL liposomal CRE-SD. Next, the cell lysates were collected using the approach described by Pengjam et al. [41]. Finally, Western blotting was performed as described in Section 2.1.8. RANKL-stimulated RAW 264.7 cells treated with CRE-SD-FREE-LIP were used as the control.

#### 2.1.10. Detecting Reactive Oxygen Species (ROS) Production in Liposomal CRE-SD-Treated RANKL-Stimulated RAW 264.7 Cells

To analyze the effect of liposomal CRE-SD on intracellular ROS production in RANKL-stimulated RAW 264.7 cells, an Oxiselect^TM^ Intracellular ROS assay kit (CellBio Lab, Inc., San Diego, CA, USA) was employed. First, 20 ng/mL RANKL-stimulated RAW 264.7 cells were incubated for 12 h with CRE-SD-FREE-LIP and liposomal CRE-SD (5, 10, and 20 µg/mL), along with 10 μM 2′,7′-dichlorofluorescein diacetate. Next, the fluorescence signal from 2′,7′-dichlorofluorescein (DCF) in each cell sample was measured at excitation and emission wavelengths of 485 and 530 nm, respectively, using a spectrofluorometer (Beckman Coulter, Inc., CA, USA). As the intracellular ROS concentration is proportional to the signal intensity of DCF, the percentage of intracellular ROS production was calculated based on the signal intensity.

#### 2.1.11. Statistical Analysis

All numerical data are expressed as the mean ± SD (*n* = 3). Statistical analyses were performed using one-way analysis of variance, the post hoc Dunnett’s test, and the Student’s *t*-test. *p*-values of <0.05 were considered to indicate statistical significance.

### 2.2. In Silico Analyses

#### 2.2.1. Retrieval of the Structures of the IκBα/p50/p65 Complex and Curcuminoids

The 3D structure of the IκBα/p50/p65 complex (Protein Data Bank (PDB) ID: 1IKN) was retrieved from RCSB PDB (https://www.rcsb.org/, accessed on 21 December 2022) [42] in PDB format. The 1IKN file comprises p65 (chain A), p50D (chain C), and IκBα (chain D) subunits. The 3D chemical structures of Cu, De, and Bis (PubChem CIDs: 969516, 5469424, and 5315472, respectively) were retrieved from the PubChem database (https://pubchem.ncbi.nlm.nih.gov/, accessed on 20 December 2022) [43] in SDF format.

#### 2.2.2. Blind Docking Simulation between the IκBα/p50/p65 Complex and Cu/De/Bis

We employed CB-Dock2, a web-based blind docking simulation tool (https://cadd.labshare.cn/cb-dock2/php/index.php, accessed on 25 January 2023) [44], for protein–ligand blind docking. After submitting the ligand and protein structure files to CB-Dock2, the chemical structures were assessed. The RDKit in CB-Dock2 adds hydrogens and partial charges to the uploaded ligand chemical structure. Furthermore, RDKit generates an initial 3D ligand conformation. Then, CB-Dock2 amends the uploaded protein structure by adding missing hydrogen atoms as well as residues and removing heteroatoms (atoms that do not belong to the protein), as well as co-crystallized water molecules. Next, a protein surface curvature-based cavity detection method, known as CurPocket, was used for binding pocket (cavity) detection. CurPocket identifies the five largest binding pockets, as well as the docking parameters (e.g., grid center coordinates, size, and volume of the docking box (search space)), of the uploaded protein. After the retrieval of the binding pocket profile and selection of the docking cavities, structure-based ligand–protein blind docking, with or without template-based ligand–protein blind docking, was executed. The former is mandatorily performed via the AutoDock Vina (version 1.1.2) algorithm, whereas the latter is conducted using FitDock when the query protein–ligand complex is homologous to the known protein–ligand complex in the BioLip database (i.e., FP2 fingerprint of ≥0.4 (high topology similarity), sequence identity of the binding pocket of ≥40%, and structure root mean squared deviation (RMSD) of the binding pocket of ≤4 Å). Finally, the blind docking information regarding the optimal binding site and binding pose (the candidate of binding mode (the orientation and conformation of the ligand that binds to the protein)) of the ligand, as well as the Vina/FitDock scores, are displayed. The superior performance of CB-Dock2 was corroborated through the 85.9% success rate of the top-ranking binding mode (RMSD of ≤2 Å) prediction, which was much higher than those achieved using other docking algorithms (CB-Dock, FitDock, MTiAutoDock, SwissDock, and COACH-D) [44].

Therefore, we conducted a blind docking simulation to explore the optimal binding site and binding pose of each curcuminoid (Cu/De/Bis) to the IκBα/p50/p65 complex. Initially, the 1IKN PDB file and the SDF files of each curcuminoid were uploaded to the CB-Dock2 web server to identify the five largest binding pockets. Then, all five cavities were used for blind docking. As the query and template ligand–protein complexes do not match, only structure-based blind docking was utilized. Finally, for each curcuminoid, the docking simulation result with the most negative Vina score, which indicates the strongest binding affinity, was analyzed to derive the binding pose, docking space (docking center and docking size), possible contact residues, and molecular interactions. The binding pose, contact residues, and molecular interactions were visualized using BIOVIA discovery studio visualizer (version 21.1.0.20298; https://discover.3ds.com/discovery-studio-visualizer-download/, accessed on 13 February 2023).

## 3. Results

### 3.1. Liposomal CRE-SD Characterization

The suspensions of CRE-SD-FREE-LIP and liposomal CRE-SD were prepared. The liposomal characteristics—liposome diameter, PDI, and zeta potential at pH 7.4—measured from both liposomal suspensions are shown in Table 1.

### 3.2. In Vitro Assessment of CRE-SD Release

To analyze the in vitro release of CRE-SD, a dialysis sac with CRE-SD and a dialysis sac with liposomal CRE-SD were studied and compared. The mass of CRE-SD released was measured and the cumulative mass percentages were calculated. As illustrated in the 70-day CRE-SD release profile (Figure 3), the proportion of the cumulative mass of released CRE-SD in CRE-SD samples increased sharply, from 2.0 ± 1.2% on day 0, to 32.1 ± 1.3% on day 30. Thereafter, it increased slightly to 36.1 ± 1.0% on day 70. In contrast, the proportion in liposomal CRE-SD samples ranged from 1.0 ± 0.8% on day 0, to 7.0 ± 1.0% on day 70, and did not dramatically change over the 70-day measurement period.

### 3.3. Optimization of RANKL Incubation Time

The optimal RANKL incubation time was investigated by detecting TRAP activity and CTSK relative expression of murine RAW 264.7 macrophages treated with CRE-SD-FREE-LIP (Figure 4A,B). In cells that were not treated with 20 ng/mL RANKL, no significant changes were found throughout the 5-day incubation period. In contrast, in the 20 ng/mL RANKL-treated cells, the TRAP activity increased significantly on days 3–5, and the CTSK relative expression was significantly elevated on days 4 and 5.

### 3.4. Optimization of Liposomal CRE-SD Concentration

CRE-SD-FREE-LIP and liposomal CRE-SD (1, 2.5, 5, 10, 20, or 30 µg/mL) were added to RAW 264.7 cells with and without 20 ng/mL RANKL treatments for 5 days. Then, we performed an MTT assay to assess the number of viable cells in each cell sample and explore the most effective liposomal CRE-SD concentration that did not induce cytotoxicity. As shown in Figure 4C, the number of viable cells significantly decreased after RANKL-stimulated RAW 264.7 cells were exposed to 30 µg/mL liposomal CRE-SD. The RANKL-stimulated and -unstimulated RAW 264.7 groups treated with up to 20 µg/mL liposomal CRE-SD had greater numbers of viable cells than did the group treated with CRE-SD-FREE-LIP, indicating the proliferative effect of liposomal CRE-SD. However, there was no statistically significant difference in the number of viable cells.

### 3.5. Inhibitory Effect of Liposomal CRE-SD on Osteoclastogenesis

To explain the liposomal CRE-SD effect on osteoclastogenesis, morphological examinations were performed, the proportions of multinucleated TRAP-positive cells and F4/80-positively stained cells were calculated, and the expression of osteoclast markers was detected. Initially, TRAP staining and a TRAP assay were used to examine the RAW 264.7 cell samples. The microscopic images indicated the existence of multinucleated TRAP-positive cells in 20 ng/mL RANKL-stimulated RAW 264.7 cell samples with CRE-SD-FREE-LIP and 20 µg/mL liposomal CRE-SD (Figure 5A and Appendix A). The percentage of multinucleated TRAP-positive cells in the 20 µg/mL liposomal CRE-SD-treated samples was significantly lower than that in the CRE-SD-FREE-LIP-treated samples (Figure 5B). Subsequently, the TRAP activities of 20 ng/mL RANKL-stimulated RAW 264.7 cell samples treated with various concentrations of liposomal CRE-SD, were measured. As shown in Figure 5C, TRAP activity was dose-dependently reduced, and it was significantly decreased in the cells treated with 20 µg/mL liposomal CRE-SD. Moreover, F4/80 staining was used to assess the proportion of macrophages in each cell sample. In Figure 5D and Appendix A, there are numerous brown patches in the RAW 264.7 cell samples treated with CRE-SD-FREE-LIP, owing to the high number of macrophages. The rarity of brown patches was apparent in the 20 ng/mL RANKL-stimulated RAW 264.7 cell samples. However, this was reversed in 20 µg/mL liposomal CRE-SD. This finding was supported by the percentage of F4/80-positively stained cells (Figure 5E), which was reduced after stimulation of CRE-SD-FREE-LIP-treated RAW 264.7 cells with 20 ng/mL RANKL. When 20 µg/mL liposomal CRE-SD was applied to the RANKL-stimulated RAW 264.7 cells, there was a significant increase in this percentage. Further, RT-PCR was performed to measure the expression of osteoclast markers (CTSK, c-Fos, and NFATc1). After the addition of 10 and 20 µg/mL liposomal CRE-SD to 20 ng/mL RANKL-stimulated RAW 264.7 cells, the fainter bands of the preceding osteoclast markers developed in a dose-dependent manner (Figure 5F and Appendix A). The expression of c-Fos significantly decreased in the cells treated with 10 µg/mL liposomal CRE-SD. Significant reductions in the expressions of CTSK, c-Fos, and NFATc1 were found in the cells treated with 20 µg/mL liposomal CRE-SD (Figure 5G–I).

### 3.6. Suppressive Effect of Liposomal CRE-SD on IκBα/p65 Phosphorylation in Osteoclastogenesis

Western blotting was utilized to detect the phosphorylation of IκBα and p65. The Western blot images of p-p65, total p65, p-IκBα, and total IκBα proteins in 20 ng/mL RANKL-stimulated RAW 264.7 cells treated with CRE-SD-FREE-LIP and liposomal CRE-SD (10 and 20 µg/mL) are illustrated in Figure 6A and Appendix A. Significant downregulations of p65 and IκBα phosphorylation were observed after exposure to liposomal CRE-SD in a dose-dependent manner (Figure 6B,C).

### 3.7. Inhibitory Effect of Liposomal CRE-SD on Nuclear Translocation and Transcriptional Activity of p-p65 in Osteoclastogenesis

To corroborate the effects of liposomal CRE-SD on nuclear translocation and the transcriptional activity of p-p65, we used Western blotting analysis to examine the p65 phosphorylation of 20 ng/mL RANKL-stimulated RAW 264.7 cells. As shown in Figure 7 and Appendix A, the p-p65/total p65 ratio decreased when 20 ng/mL RANKL-stimulated RAW 264.7 cell samples were treated with 20 µg/mL liposomal CRE-SD. In contrast, there was a significant reversal after 20 ng/mL RANKL-stimulated RAW 264.7 cell samples were treated with JSH23 and 20 µg/mL liposomal CRE-SD.

### 3.8. Inhibition of Liposomal CRE-SD on Intracellular ROS Production in Osteoclastogenesis

First, 20 ng/mL RANKL-stimulated RAW 264.7 cells were treated with CRE-SD-FREE-LIP and liposomal CRE-SD (5, 10, and 20 µg/mL) to analyze the production of intracellular ROSs. As shown in Figure 8, there was a dose-dependent reduction in the percentage of ROSs produced in the cells. Furthermore, such a proportion significantly decreased in the cells treated with 20 µg/mL liposomal CRE-SD.

### 3.9. Blind Docking Simulation between IκBα/p50/p65 Protein Complex and Cu/De/Bis

Following the completion of structure checking and cavity detection processes in CB-Dock2, information on the five largest binding pockets of the IκBα/p50/p65 protein complex was computed using CurPocket (Table 2), along with their possible contact residues (Appendix A). After the CB-Dock2 blind docking simulation between the IκBα/p50/p65 protein complex and Cu/De/Bis, we selected the docking result with the greatest negative Vina score to analyze the protein–curcuminoid interactions. The results in Table 3 demonstrate that Cu most probably interacted with binding pocket No.1 (Vina score = −8.0 kcal/mol), whereas De/Bis may have the highest binding affinity to binding pocket No. 2 (Vina score for De = −9.2 kcal/mol; Vina score for Bis = −8.8 kcal/mol). The possible contact residues of these binding pockets are shown in Table 4. There was an abundance of van der Waals forces and classical or conventional hydrogen bonds. Moreover, a moderate number of hydrophobic interactions (π, alkyl, and mixed π/alkyl (e.g., π–sigma and π–alkyl) hydrophobic interactions) were found in the Cu/De/Bis docking results. In Figure 9A–I, we present some of the intramolecular interactions within the curcuminoids, along with some favorable intermolecular curcuminoid–protein interactions, including classical and non-classical (e.g., carbon and π–donor) hydrogen bonds, electrostatic interactions (e.g., π–charge (π–cation and π–anion) interactions), alkyl and mixed π/alkyl hydrophobic interactions, and π–sulfur interactions.

### 3.10. The Inhibitory Effect of Liposomal CRE-SD on Osteoclastogenesis via the Canonical NF-κB Signaling Pathway

The in vitro modulatory effect of liposomal CRE-SD on osteoclast differentiation through the canonical NF-κB signaling pathway is shown in Figure 10. Illustrating this, liposomal CRE-SD reduced p65 and IκBα phosphorylation, as well as the nuclear translocation and transcriptional activity of p-p65 initiated by the RANK-RANKL interaction. Eventually, osteoclastogenesis was suppressed, as substantiated by the lower number of osteoclasts and decreased expression of osteoclast markers (TRAP, CTSK, c-Fos, and NFATc1), together with a higher expression of F4/80 in undifferentiated murine RAW 264.7 cells.

## 4. Discussion

Numerous drug formulations have been developed in an effort to improve the bioavailability of Cu, De, and Bis. Many clinical trials have shown that most curcuminoid formulations are safe and exhibit well-tolerated properties and excellent therapeutic efficacy with no serious adverse side effects; however, some curcuminoid formulations have been reported to exert side effects. For example, leukopenia, nausea, fatigue, and diarrhea after the administration of curcumin phosphatidylcholine and irinotecan in patients with solid tumors; anemia and hemolysis after the application of the liposomal curcumin Lipocurc™ in patients with locally advanced or metastatic tumors; and mild gastrointestinal symptoms in patients with Parkinson’s disease treated with curcumin nanomicelles have all been reported [45]. Regarding bone diseases, various curcuminoid formulations have been developed to treat osteoporosis, osteoarthritis, and RA. The oral administration of 110 mg/day curcumin with 5 mg/day alendronate prevented osteoporosis in 60 postmenopausal women by decreasing the levels of bone-specific alkaline phosphatase and C-terminal cross-linking telopeptides of type I collagen, as well as increasing the levels of osteocalcin and bone mineral density. The combined treatment could also minimize the side effects of alendronate [46,47]. The pharmaceutical advantages of safety, tolerability, and efficacy of diverse curcuminoid formulations have been demonstrated in patients with osteoarthritis [46]. The curcuminoid–turmeric matrix formulation (containing 50% total curcuminoids (41.2% Cu, 7.3% De, and 1.5% Bis), 3% essential oils, 2% protein, and 40% total carbohydrates) reduced the levels of C-reactive protein and rheumatoid factor, as well as the rate of erythrocyte sedimentation, and lowered disease activity scores/indexes in patients with RA, with no serious adverse side effects [45,46]. Owing to the above-mentioned pharmacological advantages, the development of novel curcuminoid formulations may facilitate the effective treatment of bone diseases. However, it is necessary to understand their biological effects by performing in vitro experiments and in silico analyses.

In this study, the biochemical regulation of liposomal CRE-SD in osteoclastogenesis via the canonical NF-κB pathway was clarified using both in vitro and in silico analyses. Liposomal characterization revealed that the liposome diameter was larger when CRE-SD was encapsulated in the POPC liposomes, similar to previous work showing that the solubilization of curcumin in the POPC bilayer of a liposome suspension may contribute to an increase in liposome size [38]. Several reports have mentioned the acceptable values of PDI and zeta potential to ensure that the liposome system is stable. The acceptable PDI, which is between 0 and 0.3, results in a narrow liposome size distribution and a low probability of liposome aggregate formation. An acceptable zeta potential (beyond ± 30 mV) indicates the high stability of the liposome system, on account of electrostatic activity and steric effects [48,49,50]. Here, the PDI of the liposomal CRE-SD was comparable with that of the CRE-SD-FREE-LIP. Both values are approximately the PDI of POPC liposome suspensions with and without CRE-Ter (the ternary complex of CRE prepared by dispersing PVP K30 (9% *w*/*w*) to the CRE, incorporating 2-hydroxypropyl-β-cyclodextrin (HPBCD; molar ratio = 1:1) to the CRE and performing solvent evaporation under reduced pressure, respectively; CRE-Ter contains 14% *w*/*w* curcuminoids, as demonstrated in the work of Pengjam et al. [39]. The zeta potentials, at pH 7.4, of both CRE-SD-treated and CRE-SD-FREE-LIP samples were negative, owing to the negatively charged POPC component of the liposomes. Moreover, there was a decrease in the zeta potential after treating CRE-SD in the liposome suspension. The change in such a value may be caused by the partial deprotonation of Cu/De/Bis in CRE-SD at pH 7.4 in PBS, i.e., the partial deprotonated form of the curcuminoids may provide negatively charged ions, therefore reducing the zeta potential [51]. Although the zeta potential of the liposomal CRE-SD samples met the criterion previously mentioned, the PDI may be a shortcoming that could lead to liposome aggregation. Additionally, regarding the review of Takechi-Haraya et al., the liposomal CRE-SD and CRE-SD-FREE-LIP applied in our study are polydisperse, owing to the PDI > 0.4 measured using a DLS instrument [52]. Further microscopic examination, such as through a transmission electron microscope (TEM), to assess the size distribution of liposomes with the PDI derived from DLS instrument > 0.4 was suggested by Takechi-Haraya et al. [52]. Nevertheless, conducting this microscopic observation in our study was constrained by the unavailability of electron microscopy resources. Wahyudiono et al. reported that the injection of a greater concentration of phospholipids, comprising POPC, into a medium brings about an enhancement in liposome size and PDI [53]. Previous studies by Csicsák et al. and Lee mentioned that the reduction in the size of drug particles can increase their surface area, resulting in an improvement in their dissolution and bioavailability, along with a narrow size distribution for drug particles [54,55]. In other words, decreasing the concentration of POPC utilized to produce liposomal CRE-SD may lead to reduced liposome size and PDI, as well as enhanced bioavailability. Thus, further research is needed to optimize the amount of POPC required to encapsulate the CRE-SD into liposomes, particularly before testing the pharmaceutical effectiveness of liposomal CRE-SD in in vivo studies and clinical trials.

Paradkar et al. analyzed the scanning electron microscope (SEM) images of curcumin–PVP K30 solid dispersions prepared through spray drying. The images showed spherical particles in curcumin/PVP K30 samples. The particles of the samples with higher proportions of PVP K30 (curcumin/PVP K30 ratios of 1:5–1:10) had smoother surfaces than those with curcumin/PVP K30 ratios of 1:1–1:3. Additionally, the surface of the former had concave depressions [56]. Considering the SEM images of Chhouk et al. [57], the particle sizes of PVP microspheres containing curcumin prepared via electrospraying was smaller than those of unmodified PVP particles. The incorporation of curcumin with PVP can increase conductivity and reduce particle size [57]. Machmudah et al. mentioned that modifying the surfaces of solutes with hydrophilic polymer carriers, in which PVP K30 is included, as well as decreasing the size of solutes, can increase the solubility. The former can also prevent the agglomeration of the particle product. Furthermore, the intermolecular interaction between curcumin and PVP can lead to a higher thermal stability of curcumin via carboxyl and carbonyl conjugations [58]. Altogether, the addition of PVP to curcumin may improve the solubility and stability of curcumin, resulting in better pharmacokinetic properties. Although SEM images and analytical chemistry experiments can provide more information on the morphological and chemical characteristics of the CRE-SD, which may affect its pharmacokinetic attributes, the resources of SEM and analytical chemistry instruments were unavailable in this study. Hence, further research is needed to analyze such CRE-SD characteristics.

Laouini et al. conducted TEM observation and a stability study to examine some characteristics of POPC liposomes. The micrograph showed that such liposomes were spherical. Throughout the storage period (3 months; stored at 5 ± 3 °C), there was low variation in their average size and zeta potential. In addition, neither aggregation nor sedimentation was found [59]. The in vitro examination of CRE-SD release in our study identified a lower percentage of the cumulative mass of CRE-SD lost from the dialysis sac of liposomal CRE-SD samples over 70 days, in comparison with that lost from CRE-SD samples. The slight change in this percentage from the liposomal CRE-SD samples over 70 days indicated the highly controlled and sustained release of the CRE-SD encapsulated in the POPC liposomes. Thus, utilizing POPC liposome encapsulation in the CRE-SD may lead to higher stability with a long shelf-life (70 days) compared with CRE-SD. Compared to the 60-day in vitro assessment of liposomal CRE-Ter release performed by Pengjam et al., as well as the analysis of a Cu-loaded DMPC liposomes/3D-printed tricalcium phosphate (TCP) scaffold conducted by Sarkar and Bose, the proportion of CRE-SD in the liposomal CRE-SD samples developed in this study decreased by approximately 5–10% and approximately 5–12%, respectively [39,60]. Consequently, this implies that liposome encapsulation can improve the pharmacokinetic properties of CRE-SD, and those of liposomal CRE-SD may be superior to the liposomal CRE-Ter and Cu-loaded DMPC liposomes/3D-printed TCP scaffold. However, further in vivo and clinical studies exploring the safety and efficacy of liposomal CRE-SD are required to confirm the pharmacological practicality of liposomal CRE-SD.

Although some properties of liposomal CRE-SD and the in vitro release of CRE-SD were investigated in this study, the interaction between CRE-SD and the POPC bilayers of the liposomes remains unexplored. Currently, the interactions between curcumin and POPC lipid bilayers have been illustrated. Ercan [61] used coarse-grained molecular dynamics to analyze the interactions between curcumin and lipid bilayers composed of regular POPC (containing one unsaturated chain), and those composed of regular DOPC (1,2-dioleoyl-sn-glycero-3-phosphocholine; containing two unsaturated chains) with different curcumin/lipid molar ratios. The calculated membrane thickness of regular POPC lipids was greater than that of regular DOPC lipids, in contrast to the area per lipid headgroup. Furthermore, introducing a higher amount of curcumin led to lower membrane thickness and a greater area per lipid headgroup in both regular POPC and DOPC lipids. The changes in the membrane thickness and area per lipid headgroup may be caused by the changes in the binding orientation and binding states of curcumin [61]. As the lipid compositions and the concentration of curcumin alter the size of the bilayer membrane, these factors may affect the bioavailability of curcumin. Hence, further elucidation of the interactions between CRE-SD and POPC bilayers, as well as the changes in the related parameters of POPC, is warranted to understand the effect of liposomal CRE-SD formation on its bioavailability.

We optimized the RANKL incubation time and liposomal CRE-SD concentration. Following CRE-SD-FREE-LIP and 20 ng/mL RANKL treatments for up to 5 days, the TRAP activity and the relative expression of CTSK in the RAW 264.7 cells were significantly increased on days 3–5 and days 4–5, respectively, indicating a tendency toward osteoclastogenesis within these periods. From this finding, we can determine that the optimal incubation time for liposomal-treated RANKL-stimulated RAW 264.7 cells should be at least 4 days. However, in further in vitro experiments, a 5-day incubation period after treatment with CRE-SD-FREE-LIP, liposomal CRE-SD, and/or RANKL was applied for RAW 264.7 cells, as previous reports described the attainment of osteoclastogenic stimulation using a 5-day incubation period in liposomal-treated and liposomal-untreated RANKL-stimulated RAW 264.7 cell samples [4,62]. The cell viability in 20 ng/mL RANKL-stimulated RAW 264.7 cells treated with various concentrations of liposomal CRE-SD was assessed to determine the maximum dose of liposomal CRE-SD that did not give rise to cytotoxicity, as indicated by a significant depletion in the number of viable cells. A significant reduction in viable cell number was found in 20 ng/mL RANKL-stimulated RAW 264.7 cells treated with 30 µg/mL liposomal CRE-SD. Therefore, the highest concentration of liposomal CRE-SD that did not induce cytotoxicity was 20 µg/mL, which was applied for the subsequent in vitro analyses.

Then, morphological examinations were conducted, and osteoclast marker expression was measured. The TRAP staining, TRAP assay, and RT-PCR results confirmed that 20 µg/mL liposomal CRE-SD may significantly suppress the proportion of osteoclasts (justified by that of multinucleated TRAP-positive cells) and significantly reduce the expressions of osteoclast markers (TRAP, CTSK, c-Fos, and NFATc1) in 20 ng/mL RANKL-stimulated RAW 264.7 cells. In other words, liposomal CRE-SD may hinder osteoclastogenesis. For the immunocytochemistry of F4/80, there were fewer brown patches and a lower percentage of F4/80-positively stained cells after CRE-SD-FREE-LIP-treated RAW 264.7 cells were subjected to 20 ng/mL RANKL. Their inversion was shown in RANKL-stimulated RAW 264.7 cells treated with 20 µg/mL liposomal CRE-SD. These results suggest a decrease in the proportion of murine macrophages that may be murine RAW 264.7 cells, as RANKL treatment contributes to osteoclastogenesis. In contrast, treating RANKL-stimulated RAW 264.7 cells with 20 µg/mL liposomal CRE-SD may inhibit osteoclastogenesis, as demonstrated by an increase in the proportion of undifferentiated murine RAW 264.7 macrophages. To put it simply, the immunocytochemistry analysis of F4/80 corroborates the inhibitory effect of liposomal CRE-SD on osteoclastogenesis.

Considering Western blotting, the blot images and the phosphorylated protein/total protein ratios indicated that liposomal CRE-SD may dose-dependently impede the phosphorylation of p65 and IκBα in 20 ng/mL RANKL-stimulated RAW 264.7 cells. Moreover, these results indicated that the incorporation of JSH23, and subsequent treatment with 20 µg/mL liposomal CRE-SD, could counteract the suppressed p-p65 expression in 20 ng/mL RANKL-stimulated RAW 264.7 cells. Accordingly, the findings suggest that liposomal CRE-SD can prevent the phosphorylation of p65 and IκBα, along with the nuclear translocation and transcriptional activity of p-p65 in RANKL-stimulated RAW 264.7 cells.

Previous studies have revealed that ROSs can upregulate NF-κB transcription factors, resulting in osteoclastogenesis [63,64]. Moreover, curcuminoids have an antioxidant effect, causing the removal of free radicals [65]. Our findings revealed a dose-dependent decrease in the production of intracellular ROSs after the incorporation of liposomal CRE-SD into 20 ng/mL RANKL-stimulated RAW 264.7 cells. Hence, it can be inferred that liposomal CRE-SD suppresses ROS production within osteoclasts and, hence, the cellular canonical NF-κB pathway, resulting in the inhibition of osteoclastogenesis.

Several studies have highlighted the design of liposomal curcuminoids for inhibiting in vitro osteoclast differentiation. Yeh et al. showed that Cu- and Bis-loaded soybean phosphatidylcholine (SPC) liposomes hamper osteoclastogenesis [62]. Pengjam et al. typified the impediment of liposomal CRE-Ter in osteoclast differentiation via the NF-κB and extracellular signal-regulated kinase (ERK) signaling pathways [39], through which its anti-osteoclastogenic regulation through the former supports our in vitro findings. Owing to the in vitro modulatory effect of liposomal CRE-SD on osteoclast differentiation through the canonical NF-κB signaling pathway, the liposomal CRE-SD designed in this study may be an alternative herbal medicine for preventing immoderate osteoclast differentiation. Nevertheless, to completely elucidate how liposomal CRE-SD affects the coupling of bone formation and bone resorption, future research should examine the in vitro biochemical function of liposomal CRE-SD during osteoblast differentiation. Furthermore, in vivo studies and clinical trials are required to confirm the potential of liposomal CRE-SD for the treatment of bone diseases.

In terms of the blind molecular docking simulation, we implemented the CB-Dock2 to investigate the interactions between the IκBα/p50/p65 protein complex and Cu/De/Bis. The AutoDock Vina algorithm in CB-Dock2 uses the Vina scoring function to calculate the Vina score (unit: kcal/mol), which signifies the curcumin–protein binding energy. A lower (more negative) Vina score designates higher binding affinity [66]. The molecular docking results in this study revealed low Vina scores in the binding between Cu/De/Bis and the IκBα/p50/p65 protein complex, i.e., the curcuminoids had a strong binding affinity to the protein complex. This can be attributed to high proportions of van der Waals forces, conventional hydrogen bonds, and hydrophobic interactions in the curcuminoid–protein docking mechanism, together with other minor intermolecular interactions. These implications on binding affinity are supported by a report by Saeed et al. demonstrating the low binding energies of the docking between Cu/De/Bis and the p50/p65 heterodimer complexed to IκB DNA (PDB ID: 1VKX), computed using the AutoDock program version 4.2.6 [35].

Our 3D curcuminoid–protein docking models and two-dimensional (2D) diagrams of the curcuminoid–protein interactions illustrated numerous favorable intermolecular interactions between Cu/De/Bis and the IκBα subunit of the protein complex. This might substantiate that these interactions bring about the inhibitory effects of curcuminoids in liposomal CRE-SD on IκBα phosphorylation, proven in our in vitro experiments. Considering the p65-p50 dimer in the protein complex (PDB ID: 1IKN), the N-terminal Rel homology region or Rel homology domain (RHR or RHD; positions 19–315 of p65 and positions 39–368 of p50) consists of functional sequences, i.e., a DNA-binding subdomain (positions 19–185 of p65 and positions 39–238 of p50), a dimerization subdomain (positions 191–291 of p65 and positions 245–350 of p50), and a nuclear localization sequence (NLS; positions 301–304 of p65 and positions 361–363 and 365 of p50) [27]. Many of the favorable intermolecular interactions shown in our docking output indicated the binding of the curcuminoids to such functional sequences. Cu binds to the dimerization subdomain of p50 at THR256 via conventional hydrogen bonding, and at LEU346 via π–alkyl interactions. De binds to the DNA-binding subdomain of p65 at GLN29 via π–donor hydrogen bonds, and at ARG30 via conventional hydrogen bonding. It also binds to the dimerization subdomain of p65 at LYS221 via alkyl, π–alkyl, π–donor, and π–cation interactions, and at VAL244 via conventional hydrogen bonding. Bis binds to the DNA-binding subdomain of p65 at GLN29, LYS79, and HIS181 via conventional hydrogen bonding, at ARG158 via π–cation interaction, and at PRO182 via carbon–hydrogen bonds and π–alkyl interactions. Furthermore, it binds to the dimerization subdomain of p65 at LYS221 via conventional hydrogen bonding. These findings suggest that the interactions of the curcuminoids in liposomal CRE-SD to DNA binding and dimerization subdomains may cause an impediment of p65-p50 dimerization and the binding of the p65-p50 dimer to nuclear DNA. Our CB-Dock2 docking revealed that there were no interactions between the curcuminoids and the NLS of the p65 and p50 subunits, contradicting the AutoDock version 4.0 docking simulation results of curcumin, as reported by Cheemanapalli et al. [34]. A plausible cause for this is a difference in the binding pocket prediction methods. The CurPocket in CB-Dock2 retrieves the binding pocket candidates by screening the clusters of concave surfaces, whereas the Computed Atlas of Surface Topography of Proteins (CASTp) program, which Cheemanapalli et al. utilized for identifying protein active sites, creates the binding pocket profile using the alpha shape method [34,44,67,68]. In this study, owing to the absence of the NLS residues in the top-five binding pocket information derived from the CurPocket cavity detection process, the NLS residues were not implemented in CB-Dock2 blind docking. The in vitro experiments by Riedlinger et al. mentioned that monomeric p65 can be rapidly degraded and ineffectively translocated to the nucleus [69]. For this reason, our in silico findings implied that the suppression of p65-p50 dimerization and the binding between p65 and DNA arising from the curcuminoid intermolecular interactions may reduce the stability of p65 and, consequently, inhibit its phosphorylation, nuclear translocation, and DNA binding. Although this inference corroborates our in vitro results, it cannot fully confirm the hindrance of liposomal CRE-SD on the canonical NF-κB signaling pathway during osteoclastogenesis, owing to the lack of Western blotting data concerning the regulatory effect of liposomal CRE-SD on p50 phosphorylation.

## 5. Conclusions

In summary, we applied liposomal encapsulation and PVP K30 solid dispersion to a CRE containing Cu, De, and Bis to acquire liposomal CRE-SD. The in vitro experiments corroborated the potential of liposomal CRE-SD to inhibit osteoclastogenesis (the differentiation of RANKL-stimulated murine RAW 264.7 macrophages) via the canonical NF-κB signaling pathway. Furthermore, the molecular docking simulation demonstrated that the intermolecular interactions between the curcuminoids in liposomal CRE-SD and the IκBα/p50/p65 protein complex may contribute to such suppressive regulation. These findings manifested that liposomal CRE-SD is a promising herbal candidate for bone disease treatment; nevertheless, clarification of its pharmacological effects on osteoblasts, animal models, and patients with bone disease is needed.

## Figures and Tables

**Figure 1 pharmaceutics-15-02248-f001:**
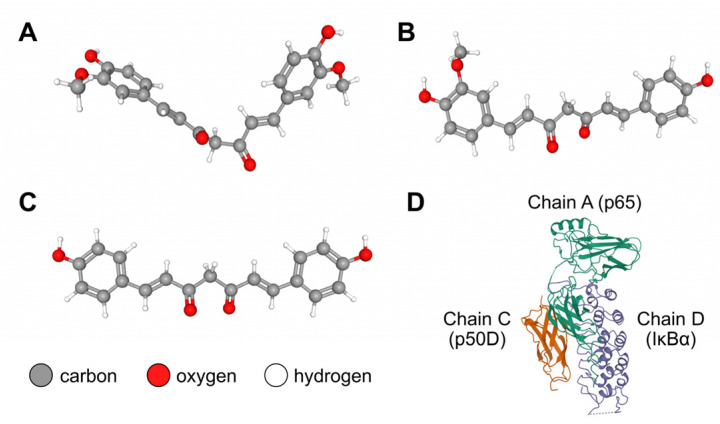
Three-dimensional chemical structures of (**A**) curcumin (Cu; PubChem compound identifier (CID): 969516); (**B**) demethoxycurcumin (De; PubChem CID: 5469424); (**C**) bisdemethoxycurcumin (Bis; PubChem CID: 5315472); and (**D**) IκBα/p50/p65 protein complex (Protein Data Bank (PDB) ID: 1IKN, consisting of p65 (chain A), p50D (chain C), and IκBα (chain D) subunits). The structures of the curcuminoid substances (Cu, De, and Bis) and the protein complex were retrieved from the PubChem and PDB databases, respectively.

**Figure 2 pharmaceutics-15-02248-f002:**
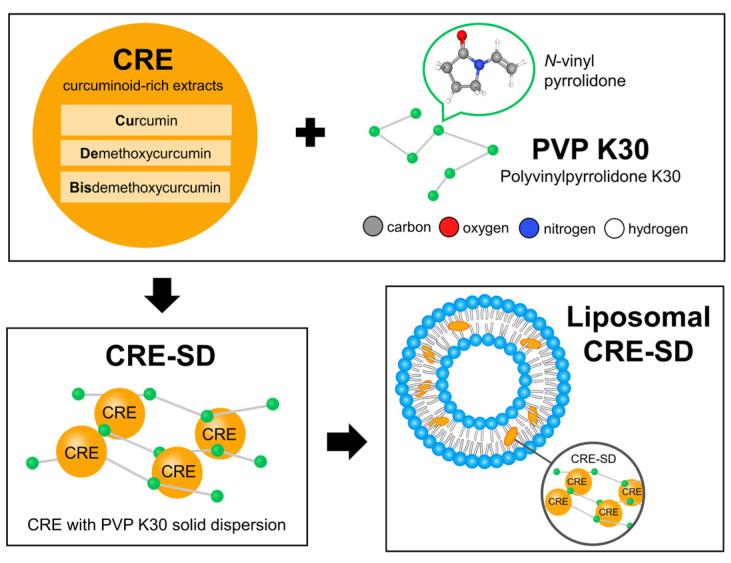
Liposomal CRE-SD preparation process. The dried powders of *Curcuma longa* rhizomes were applied to derive a curcuminoid-rich extract (CRE) comprising curcumin, demethoxycurcumin, and bisdemethoxycurcumin. To obtain CRE-SD with 7% *w*/*w* of curcuminoids, the resulting CRE was modified with polyvinylpyrrolidone K30 (PVP K30)—the synthetic polymer produced from the monomer termed *N*-vinyl pyrrolidone—using the solid dispersion technique. To obtain liposomal CRE-SD, the liposomal encapsulation method was subsequently used to interpose the CRE-SD in the lipid bilayer of the POPC liposomes.

**Figure 3 pharmaceutics-15-02248-f003:**
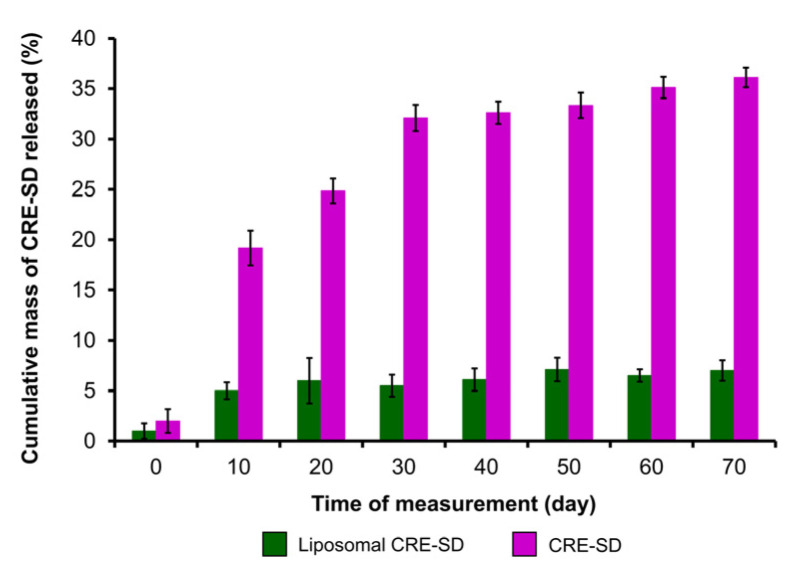
In vitro assessment of CRE-SD release after liposomal encapsulation. The percentages of the cumulative mass of CRE-SD released in liposomal CRE-SD and CRE-SD samples were measured from day 0 to 70, every 10 days (*n* = 3).

**Figure 4 pharmaceutics-15-02248-f004:**
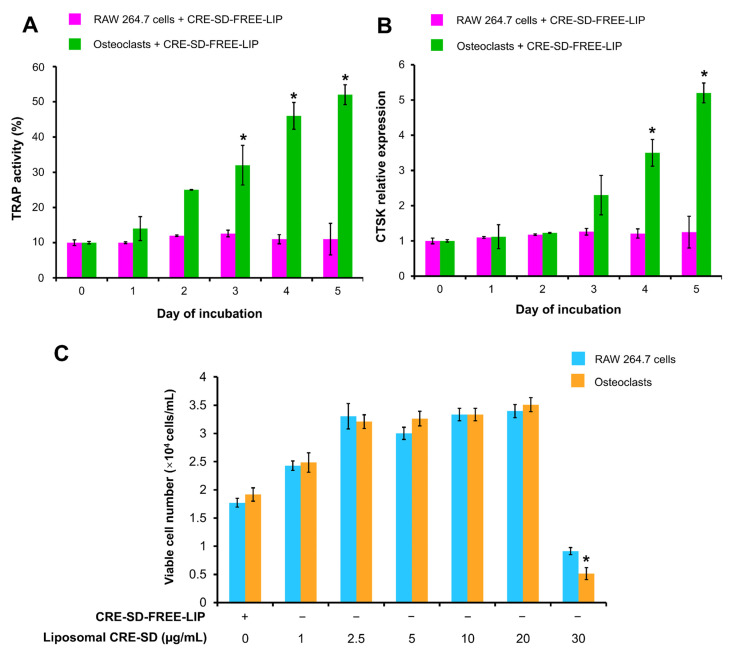
Optimization of RANKL incubation time and liposomal CRE-SD concentration. In the first study, RAW 264.7 cells were treated with 20 ng/mL RANKL and CRE-SD-free POPC liposomes (CRE-SD-FREE-LIP) for 1, 2, 3, 4, and 5 days to detect (**A**) TRAP activity and (**B**) the relative expression of cathepsin K (CTSK). Statistically significant differences in cell samples between any incubation day and incubation day 0 are indicated with an asterisk (*). In the second study, RAW 264.7 cells were treated with CRE-SD-FREE-LIP and liposomal CRE-SD (1, 2.5, 5, 10, 20, and 30 µg/mL) and incubated for 5 days with and without 20 ng/mL RANKL to estimate (**C**) the number of viable cells. Statistical significance of the difference between the corresponding liposomal CRE-SD-treated and CRE-SD-FREE-LIP-treated samples is denoted as * *p* < 0.05, *n* = 3.

**Figure 5 pharmaceutics-15-02248-f005:**
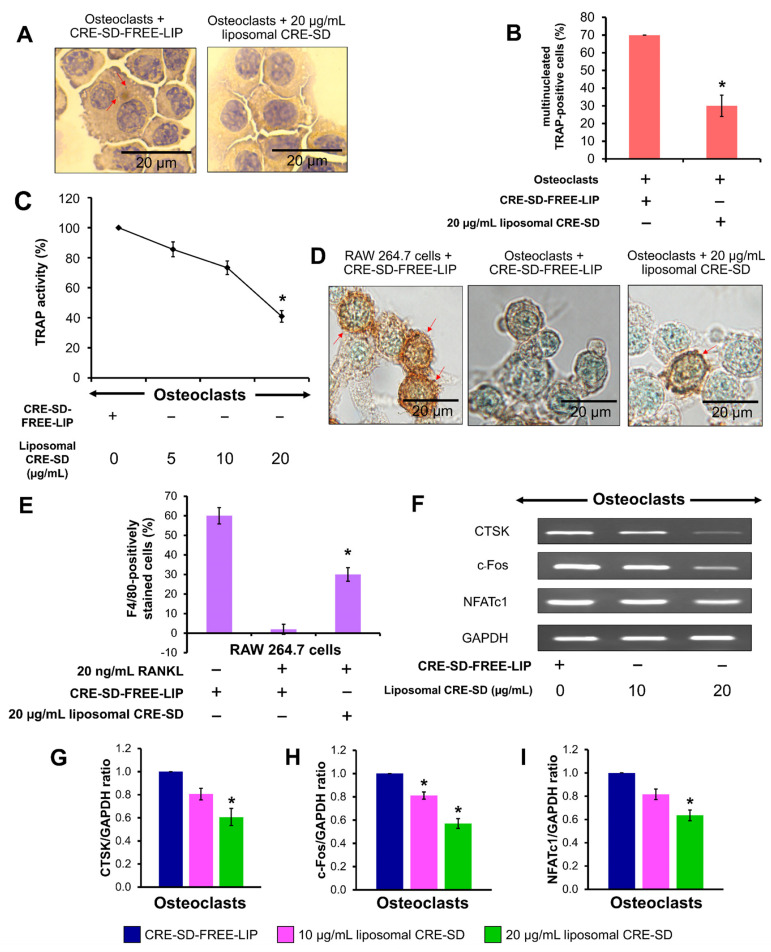
Osteoclastogenic examination. (**A**) Morphological examination after TRAP staining. First, 20 ng/mL RANKL-stimulated RAW 264.7 cells treated with CRE-SD-free POPC liposomes (CRE-SD-FREE-LIP) and 20 µg/mL liposomal CRE-SD were stained. The arrows indicate stained granules. (**B**) Percentage of multinucleated TRAP-positive cells in 20 ng/mL RANKL-stimulated RAW 264.7 cells treated with CRE-SD-FREE-LIP and 20 µg/mL liposomal CRE-SD. (**C**) TRAP activity of 20 ng/mL RANKL-stimulated RAW 264.7 cells. The CRE-SD-FREE-LIP and liposomal CRE-SD (5, 10, and 20 µg/mL) were incorporated into the cell samples. (**D**) Morphological examination after immunocytochemistry analysis of F4/80 was performed. RANKL-unstimulated RAW 264.7 cells treated with CRE-SD-FREE-LIP and 20 ng/mL RANKL-stimulated RAW 264.7 cells treated with CRE-SD-FREE-LIP and 20 µg/mL liposomal CRE-SD were examined. Arrows indicate F4/80-positively stained cells. (**E**) The percentage of F4/80-positively stained cells. RANKL-unstimulated RAW 264.7 cells treated with CRE-SD-FREE-LIP and 20 ng/mL RANKL-stimulated RAW 264.7 cells treated with CRE-SD-FREE-LIP and 20 µg/mL liposomal CRE-SD were utilized. (**F**) RT–PCR analysis of osteoclast markers (CTSK, c-Fos, and NFATc1); 20 ng/mL RANKL-stimulated RAW 264.7 cells were treated with CRE-SD-FREE-LIP and liposomal CRE-SD (10 and 20 µg/mL). (**G**) CTSK/GAPDH ratio. (**H**) c-Fos/GAPDH ratio. (**I**) NFATc1/GAPDH ratio. Statistically significant differences between the corresponding samples and the control are denoted as * *p* < 0.05, *n* = 3.

**Figure 6 pharmaceutics-15-02248-f006:**
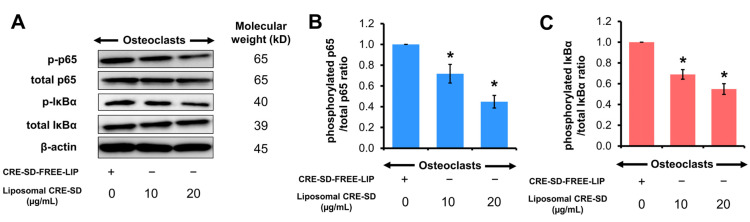
Regulatory effect of liposomal CRE-SD on p65 and IκBα phosphorylation. (**A**) Western blots of phosphorylated p65 (p-p65), total p65, phosphorylated IκBα (p-IκBα), and total IκBα after exposure of 20 ng/mL RANKL-stimulated RAW 264.7 cells to liposomal CRE-SD. (**B**) The p-p65/total p65 ratio after exposure of 20 ng/mL RANKL-stimulated RAW 264.7 cells to liposomal CRE-SD. (**C**) The p-IκBα/total IκBα ratio after 20 ng/mL RANKL-stimulated RAW 264.7 cells were exposed to liposomal CRE-SD. Statistically significant differences between the corresponding liposomal CRE-SD-treated and CRE-SD-FREE-LIP-treated samples are denoted as * *p* < 0.05, *n* = 3.

**Figure 7 pharmaceutics-15-02248-f007:**
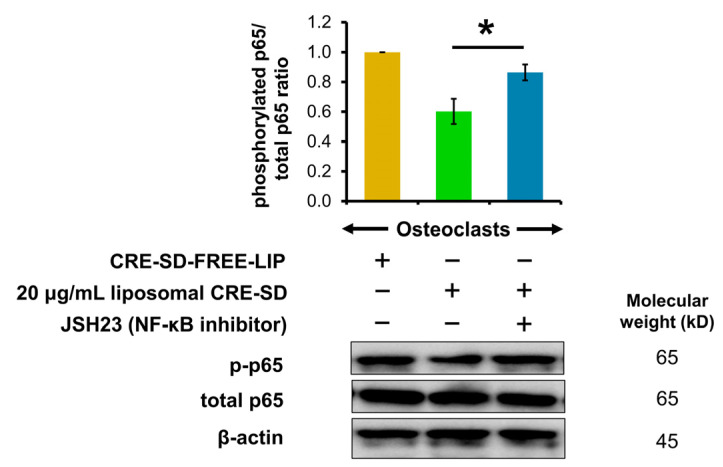
Regulatory effect of liposomal CRE-SD on p65 phosphorylation after incorporating JSH23. The Western blots and p-p65/total p65 ratio after subjecting 20 ng/mL RANKL-stimulated RAW 264.7 cells to 20 µg/mL liposomal CRE-SD and JSH23 are shown. Statistical significance of the difference between the samples treated with 20 µg/mL liposomal CRE-SD and JSH23 and the samples treated with 20 µg/mL liposomal CRE-SD is denoted as * *p* < 0.05, *n* = 3.

**Figure 8 pharmaceutics-15-02248-f008:**
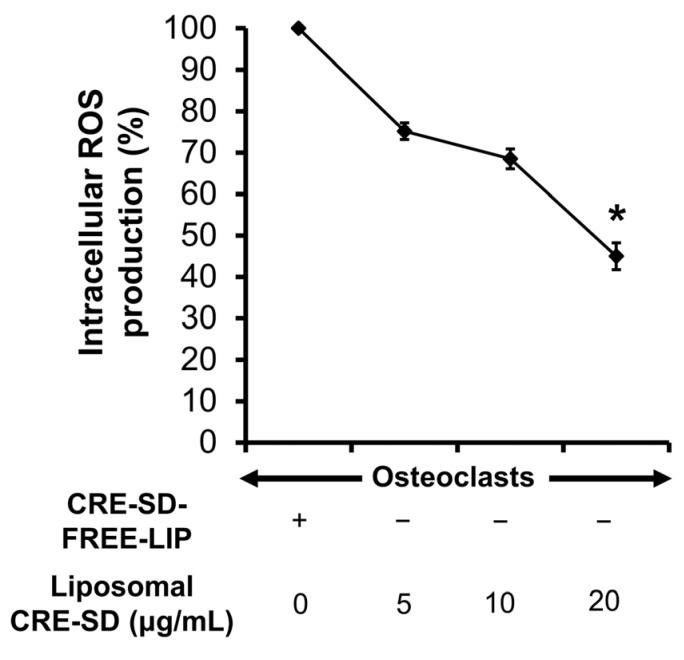
Percentage of intracellular reactive oxygen species (ROSs) produced in 20 ng/mL RANKL-stimulated RAW 264.7 cells. The cells were treated with CRE-SD-FREE-LIP and liposomal CRE-SD (5, 10, and 20 µg/mL). Statistical significance of the difference between the corresponding liposomal CRE-SD-treated and CRE-SD-FREE-LIP-treated samples is denoted as * *p* < 0.05, *n* = 3.

**Figure 9 pharmaceutics-15-02248-f009:**
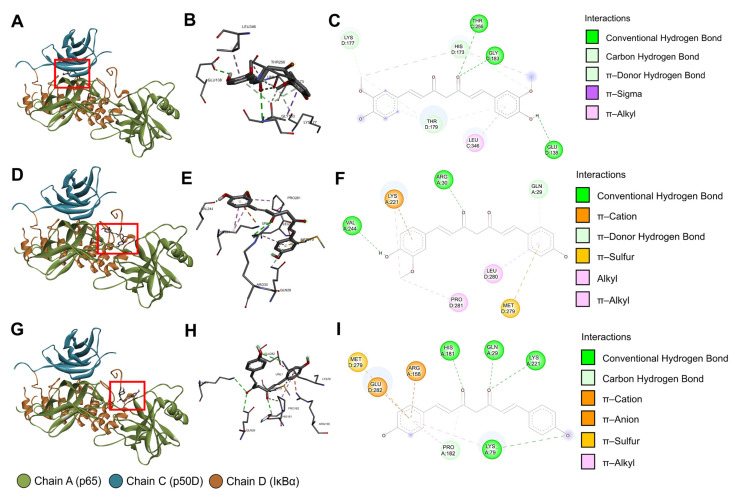
CB-Dock2 blind docking results of the IκBα/p50/p65 protein complex and curcumin/demethoxycurcumin/bisdemethoxycurcumin. (**A**) The three-dimensional (3D) binding pose of curcumin. (**B**) The 3D interactions between the IκBα/p50/p65 protein complex and curcumin. (**C**) The two-dimensional (2D) diagram of the interactions between the IκBα/p50/p65 protein complex and curcumin. (**D**) The 3D binding pose of demethoxycurcumin. (**E**) The 3D interactions between the IκBα/p50/p65 protein complex and demethoxycurcumin. (**F**) The 2D diagram of the interactions between the IκBα/p50/p65 protein complex and demethoxycurcumin. (**G**) The 3D binding pose of bisdemethoxycurcumin. (**H**) The 3D interactions between the IκBα/p50/p65 protein complex and bisdemethoxycurcumin. (**I**) The 2D diagram of the interactions between the IκBα/p50/p65 protein complex and bisdemethoxycurcumin. The red rectangles indicate the locations of the curcuminoid molecules, and the light green spheres in the 3D interactions indicate intramolecular interactions within the curcuminoid molecules. In the 2D diagrams and 3D interactions, the colored dashed lines linking the protein complex and curcuminoid denote favorable intermolecular interactions. The three-letter amino acid codes and positions of the contact residues are shown in the 3D interactions. The colored circles in the 2D diagrams denote the three-letter amino acid codes, occupying protein chains, and amino acid positions of the contact residues. The light blue circles surrounding the preceding circles are solvent-accessible surfaces. All figures were constructed using BIOVIA discovery studio visualizer.

**Figure 10 pharmaceutics-15-02248-f010:**
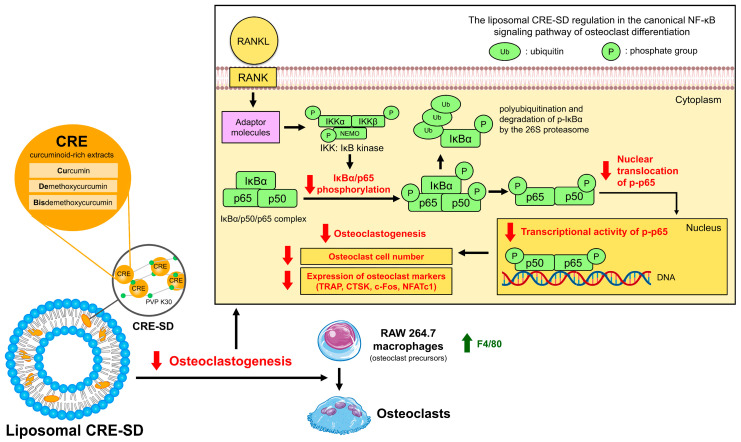
The inhibitory effect of liposomal CRE-SD on osteoclastogenesis via the canonical NF-κB signaling pathway.

**Table 1 pharmaceutics-15-02248-t001:** Liposomal characteristics.

Liposome Suspension	Liposome Diameter (nm)	Polydispersity Index (PDI)	Zeta Potential at pH 7.4 (mV)
CRE-SD-FREE-LIP	328.0 ± 14.5	0.45 ± 0.11	−22.22 ± 0.51
Liposomal CRE-SD	380.1 ± 20.5	0.43 ± 0.13	−31.00 ± 0.48

**Table 2 pharmaceutics-15-02248-t002:** CurPocket-based information on the five largest binding pockets of IκBα/p50/p65 protein complex.

Binding Pocket	Cavity Volume (Å^3^)	Center (x, y, z)	Cavity Size (x, y, z)
1	5316	21.2, 32.5, 4.8	30, 30, 30
2	2633	49.8, 32.5, 31.0	21, 22, 29
3	1341	28.0, 30.0, 26.4	15, 20, 17
4	1291	44.2, 16.7, 44.0	21, 15, 24
5	654	42.0, 25.7, 49.6	18, 16, 14

**Table 3 pharmaceutics-15-02248-t003:** The IκBα/p50/p65 binding pockets with the greatest negative Vina scores after blind docking with curcuminoids.

Ligand	Binding Pocket	Vina Score (kcal/mol)	Cavity Volume (Å^3^)	Docking Center (x, y, z)	Docking Size (x, y, z)
Curcumin	1	−8.0	5316	21, 33, 5	35, 35, 35
Demethoxycurcumin	2	−9.2	2633	50, 33, 31	27, 27, 34
Bisdemethoxycurcumin	2	−8.8	2633	50, 33, 31	26, 26, 34

**Table 4 pharmaceutics-15-02248-t004:** Possible contact residues of the IκBα/p50/p65 binding pockets in Table 3.

Ligand	Contact Residues
Curcumin	Chain C: THR256 ALA257 PRO324 PRO344 PHE345 LEU346 Chain D: GLU138 ARG140 GLY144 HIS173 LYS177 ALA178 THR179 ASN180 TYR181 ASN182 GLY183 THR185 GLN212 PRO214
Demethoxycurcumin	Chain A: GLN29 ARG30 LYS79 HIS181 PRO182 PHE184 VAL219 GLN220 LYS221 GLU222 GLN241 VAL244 HIS245 ARG246 GLN247 Chain D: TYR251 TRP258 MET279 LEU280 PRO281 GLU282 SER283 GLU287 SER288
Bisdemethoxycurcumin	Chain A: LYS221 GLU222 VAL244 HIS245 ARG246 GLN247 Chain C: ASN247 LYS249 VAL251 ASP271 Chain D: TYR248 GLN249 GLY250 TYR251 PRO281 GLU282 SER283 GLU284 GLU287 SER288 TYR289 ASP290 THR291 GLU292 SER293

## Data Availability

The data generated and analyzed during the current study are available from the corresponding author upon reasonable request.

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
