# Peer review of "Modified Curcuminoid-Rich Extract Liposomal CRE-SDInhibits Osteoclastogenesis via the Canonical NF-κB Signaling Pathway"

_pharmaceutics, 2023, doi:10.3390/pharmaceutics15092248_

Round 1
Reviewer 1 Report (Previous Reviewer 1)
The manuscript has been revised and the most comments were addressed. However, some questions/comments in the previous manuscript were not answered.
1. The terms of the liposomes were not synchronized. In table 1, two liposomes are named as “CRE-SD-free POPC liposomes” and “CRE-SD-treated POPC liposomes”, respectively. However, two liposomes in Figure 3 are named “liposome-encapsulated and liposome-unencapsulated CRE-SD”, respectively. In the Discussion section, CRE-SD-treated liposome suspension and the CRE-SD-FREE-LIP suspension were described. If there were only two samples investigated in this paper, why the authors can not name them in a simple way? Nevertheless, what does the meaning of “liposome-unencapsulated CRE-SD”? If it is free drug (CRE-SD), that sample should be “CRE-SD”. There is nothing related to liposome-unencapsulated. Besides, the authors should define the term “CRE-Ter” clearly, instead of simply show its abbreviation in the manuscript.
2. Concerning the liposomes investigated in this paper, CRE-SD was mixed with PVP, prior to the encapsulation into the liposome. Although the authors cited their previous publications in the manuscript to show their experience on the liposomes, the encapsulation of CRE-SD/PVP into the liposome might not be published. Therefore, the authors are advised to provide details of the characterizations, for instance the particle size and the morphology of CRE-SD/PVP. Its size could potentially affect the encapsulation efficiency of CRE-SD inside the liposome. Yet the released CRE in the macrophages can also affect the signaling pathway of osteoclastogenesis.
The revised manuscript has reached a certain quality for the publication. The remained comments are provided to improve the scientific merits of the paper.
Author Response
Please see the attachment.

Reviewer 2 Report (New Reviewer)
The following minor issues should be clarified:
1) Page 4, line 157-158: How was the % content of curcuminoids determined? These % results are given under "Methods" section and it is therefore not clear how this was measured.
2) Pag 5, lines 190-191: Please clearly explain what is meant by "liposome-unencapsulated CRE-SD"? Is this the same as later given in the "Results" section (p. 9, line 381) as "CRE-SD-FREE-LIP"?
Author Response
Please see the attachment.

Reviewer 3 Report (New Reviewer)
I will suggest the following changes or addition :
1) In 2.1.2- For in-vitro release the temperature should be written in ±, "" the flask, maintained at 37°C""
2) Zeta potential should be recorded with pH. Give the value of pH.
3) The entrapment and morphology of liposomes are important parts when it comes to material-related studies.
4) cut short the discussion, it is so long and vague. stay to the point. Reader interest must be maintained at all point,
The language can be improved further
Author Response
Please see the attachment.

This manuscript is a resubmission of an earlier submission. The following is a list of the peer review reports and author responses from that submission.
Round 1
Reviewer 1 Report
The manuscript entitled "Liposomal CRE-SD, a modified curcuminoid-rich extract, inhibits osteoclastogenesis via the canonical NF-κB signaling pathway, as substantiated by in vitro experiments and blind docking simulation " reports a liposomal dosage form of curcuminoid for combating osteoclastogenesis. The molecular mechanism to suppress the osteoclast activity was also explored. Here are comments and questions for authors:
1. In the abstract, the sentence describing pharmacokinetic attributes is ambiguous. There are no pharmacokinetic experiments shown in the manuscript. In addition, what are superior pharmacokinetic properties as mentioned in the abstract? I can only figure out the data in the manuscript is associated with the suppression of the osteoclast in the presence of curcuminoid-encapsulated liposomes.
2. In the materials and methods, the authors didn’t describe the preparation of liposomal curcuminoid. Even the procedures of the liposome preparation have been published in the references, a concise and correct description is important to the readers of this paper.
3. Concerning the dosage form design, how a single PC can produce stable liposomes? Alternatively, how the addition of CRE-SD along with PC can form stable liposomes?
4. Reference 39 demonstrated similar results of CRE-SD liposome, such as the signaling pathway. It seems that both of the reference and the present study examined the same cellular responses of CRE-SD liposome. The authors are advised to describe the differences between these two studies.
5. In the experimental section, “F4/80 expression” is unknown to the readers. Since the term is first appeared in the manuscript, the purpose of the expression (and staining) should be described.
6. Regarding CRE-SD-encapsulated liposome or CRE-free liposome, the authors are advised to synchronize the terms of CRE-SD liposomes throughout the manuscript. To name a few, what's the difference between the samples in table 1 and figure 3?
7. In the results, figure 4 showed the viable cell number versus the concentrations of liposomal CRE-SD. why the cell numbers for the concentrations (1, 2.5, 5, 10, 20) were more than that of control (zero liposomal CRE-SD)? Did it mean a proliferative effect?
8. Although the in vitro modulatory effect of liposomal CRE-SD on osteoclast differentiation was shown in the present study, the manuscript has been found contradicted results. Figure 3 shows that CRE-SD encapsulated in the liposome was only released less than 5% in the first 5 days. However, the in vitro modulatory effect was significant. The general purpose of drug carrier is to release right amount of drug at the right release timing. The route of drug (CRE-SD) to reach the site of action (diseased lesion) should be considered in the design of dosage form (injection, oral or other application). For injection application, the payload (drug) should be released in few hours.
All in all, the present study reports quality data in terms of cellular signaling pathway. However, the design of dosage form is not well described in the manuscript. The manuscript should be massively revised to address the comments mentioned above.
Reviewer 2 Report
Authors are suggested to resolve these issues in the manuscript before acceptance.
1. Title of the manuscript is too long. I suggest to modify informative title.
2. What is the purity of curcuminoid-rich extract (CRE)?
3. Line 165-169, in vitro release studies need to rewrite clearly.
4. Are liposomes stable? Natural-based liposomes have highly permeable and low stable properties. Suggested to perform stability studies.
5. Is PDI value (0.45 ± 0.11) monodisperse. Give suitable references.
6. Authors are suggested to add TEM images of liposomes.
Reviewer 3 Report
The authors discuss how liposomal CRE-SD, a modified curcuminoid-rich extract, can inhibit osteoclastogenesis (the formation of osteoclasts, which break down bone tissue) by suppressing the canonical NF-κB signalling pathway. In vitro experiments and blind docking simulations were conducted to support the findings. The article presents the possibility of treating bone diseases with excessive osteoclastogenesis using liposomal CRE-SD.
The manuscript is well-written and the data is very well presented.
A few comments need to be addressed before acceptance:
- the methods session is missing details. For example, how the CRE-SD is encapsulated within liposomes. The authors refer to previous literature but it would be better to state the general procedure in the manuscript for clarity.
- The paper does not contain any specific information about the potential side effects or drawbacks to using liposomal CRE-SD to treat bone diseases.
- More insight should be given on the potential advantages of using curcuminoids as a therapeutic agent for treating bone diseases.
- More information should be given on how the bioavailability of liposomal CRE-SD could be affected by various factors such as lipid composition and size distribution of the liposomes used.
The manuscript is well-written and the data is very well presented.
Reviewer 4 Report
Authors would like to prepare a curcumincid-rich extract loaded liposome and to prove the inhibiting osteoclastogenesis effect of the the new combinations and dosage forms. it seems kind of meaningful.While the experimental design and results couldn't make a orresponding conclusions. 1. Authors mentioned three main curcuminoids and also showed their 3D chemical structures in Figure1, while the components and contents of the curcumincid-rich extract were not provided. 2. We know that pharmacokinetic disadvantage is the big problem of curcuminoids.In the abstract, the author said "In vitro release assessment revealed a lower cumulative mass percentage of CRE-SD was released from liposomal encapsulated CRE-SD compared with liposomal unencapsulated CRE-SD" while author made " indicating the amelioration of pharmacokinetic attributes" conclusion. How does it manifest? 3.It lacks the in vivo pharmacokinetic research. 4.Figure 5D, Incorrect expression 5.Figure 6 and Figure 7, the data is not consistent. 6.The units are missing in ordinate in many figures.